:ﾟ. PLOS | ONE

# Arresting visuospatial stimulation is insufficient to disrupt analogue traumatic intrusions

Thomas Meyer[1,2,3]*, Chris R. Brewin[2], John A. King[2], Desiree Nijmeijer[1], Marcella L. Woud[4], Eni S. Becker[1]

**1** Behavioural Science Institute, Radboud University Nijmegen, Nijmegen, The Netherlands, **2** Department of Clinical, Educational and Health Psychology, University College London, London, United Kingdom, **3** Institute of Psychology, University of Münster, Münster, Germany, **4** Mental Health Research and Treatment Center, Department of Psychology, Ruhr-Universität Bochum, Bochum, Germany

* t.meyer@bsi.ru.nl

**Data Availability Statement:** The analysed dataset can be obtained via the Open Science Framework using the following link: https://osf.io/f38cd/.

**Funding:** The authors received no specific funding for this work.

## Abstract

Intrusive memories are a core symptom of Post-Traumatic Stress Disorder (PTSD). A growing body of analogue studies using trauma films suggest that carrying out specific demanding tasks (e.g., playing the video game Tetris, pattern tapping) after the analogue trauma can reduce intrusive memories. To examine the mechanism behind this effect, we tested whether mere engagement with attention-grabbing and interesting visual stimuli disrupts intrusive memories, and whether this depends on working memory resources and/or the concurrent activation of trauma film memories. In a total sample of 234 healthy participants, we compared no-task control conditions to a perceptual rating task with visually arresting video clips (i.e., non-emotional, complex, moving displays), to a less arresting task with non-moving, blurred pictures (Study 1), and to more demanding imagery tasks with and without repetitive reminders of the trauma film (Study 2). Generally, we found moderate to strong evidence that none of the conditions lead to differences in intrusion frequency. Moreover, our data suggest that intrusive memories were neither related to individual differences in working memory capacity (i.e., operation span performance; Study 1), nor to the degree of engagement with a visuospatial task (i.e., one-week recognition performance; Study 2). Taken together, our findings suggest that the boundary conditions for successful interference with traumatic intrusions may be more complex and subtle than assumed. Future studies may want to test the role of prediction errors during (re-)consolidation, deliberate efforts to suppress thoughts, or the compatibility of the task demands with the individual's skills.

## Introduction

A large number of individuals are exposed to potentially traumatic events at some point in their life, such as war, life-threatening accidents, or being held captive [1]. Many trauma victims continue to suffer from recurrent memories that intrude into consciousness in the form of flashbacks, or intrusions. Intrusive memories are a core symptom of post-traumatic stress

**Competing interests:** The authors have declared that no competing interests exist.

disorder (PTSD), a disorder with devastating consequences for functioning and well-being [2, 3]. This makes intrusive memories a key target in efforts to develop early interventions for trauma victims.

Various analogue studies with trauma films have shown that carrying out specific demanding tasks (e.g., playing the video game Tetris, pattern tapping, N-back tasks) during or after film viewing can reduce involuntary film memories in the following weeks (for reviews, see [4, 5, 6]). For instance, studies have found reductions in intrusive memories in participants who played Tetris immediately after a viewing a trauma film [7], after 30 min (e.g., [8, 9]), or after 4 h [10] The fact that tasks administered *after* film viewing can be effective suggests that traumatic memories are sensitive to disruption while they undergo the processes of consolidation or reconsolidation/re-encoding (i.e., during and shortly after formation or activation of the memory trace), in line with the view that newly formed memories are prone to retroactive interference for up to several hours [11, 12]. In parallel to these analogue studies, accumulating evidence suggests that already consolidated distressing autobiographical memories can lose their vividness and emotionality when the individual repeatedly executes guided eye movements whilst remembering [13, 14]. Based on these findings, many clinicians and researchers are hopeful that a disruptive effect on intrusive trauma memories can be exploited in effective treatment and preventive interventions for trauma victims (e.g., [15, 16, 17]).

To date, however, the mechanisms by which dual tasks administered after trauma exposure can reduce intrusive trauma memories remain ill-understood. A widely held view is that interference with intrusive memories depends on a task's ability to tap into limited cognitive resources that are required for trauma-related imagery and for the consolidation of the traumatic memories [4, 5]. Although there have been debates about the role of modality-specific working memory components that may be linked more specifically to imagery (e.g., the visuo-spatial sketchpad; [18]), many scholars tend to concur that imposing high levels of working memory load seems critical for the dual-task interference effects (e.g., [8, 14, 19, 20]). Indeed, several research lines highlight the role of working memory resources in the development of intrusive memories. For instance, a higher working memory capacity (WMC)–i.e. a larger amount of goal-relevant information that can be held active in working memory in the presence of distractors [21]–has been linked to fewer intrusive memories of negative personal experiences [22], or a better ability to suppress intrusive thoughts [23, 24]. Moreover, some studies suggest that WMC modulates the benefits of dual-task interference on the development of intrusive memories. For instance, Gunter and Bodner [25] found that three different visual or spatial tasks were able to reduce memory vividness and emotionality, and more strongly so in participants who had a low WMC, as measured by means of reading span.

However, some studies did not find an association between intrusive memories and WMC (e.g., as measured on an operation span task; [26], cf. [27]), whereas others could not find a dose-response relationship between the cognitive load of a dual task and benefits in the reduction of intrusions [28]. In addition, some studies have failed to replicate effects of visuospatial tasks on intrusions, or to link WM taxation to intrusion number and quality. For instance, Asselbergs et al. [29] found no experimental effects for two novel visuospatial gaming apps that required spatial planning and decision making. As a potential explanation, both of these apps may have required fewer visuospatial resources than tasks that have been shown to reduce intrusions such as Tetris. However, the authors matched one of their apps to Tetris in terms of its ability to interfere with a simple dual reaction time task, meaning that the null findings cannot be attributed to a general lack of WM taxation. In a similar vein, van Schie et al. [30] found that performing eye movements or counting whilst remembering the hotspot of a traumatic film reduces intrusive memories, but this effect could not be replicated in two out of three experiments. In addition, in all three experiments, they found no correlation between WM

taxation (i.e., interference with dual reaction time task) and a reduction in intrusion vividness or emotionality. Together, these findings point out that there are disparities in the effectiveness of visuospatial tasks to reduce intrusive memories that cannot be entirely attributed to WM taxation and/or individual differences in WMC.

An alternative possibility to the working memory hypothesis is that the beneficial dual task effects are driven by attentional redirection. Accordingly, any task that attracts attention and diverts it away from the traumatic experience might be able to interfere with the consolidation of these distressing memories. Although some studies have addressed this idea (also coined "distraction" hypothesis; e.g., [31]), they compared demanding tasks differing in modality, rather than manipulating attention-grabbing characteristics within the same modality. Still, suggestive evidence comes from Kavanagh et al. [14], who found decreased vividness and emotionality for emotional memories when participants were stimulated with "visual noise" (i.e., a screen of flickering squares) during recall, as compared to no-task control. Although this reduction was not as effective as a more taxing eye-movement condition, it provides tentative evidence that a relatively passive viewing condition can reduce the emotional impact of traumatic memories. Notably, this would have important implications for theoretical models of intrusive memories, as well as practical implications for cheap and low-demanding interventions for trauma victims (e.g., in emergency care settings).

With these considerations in mind, we explored in two studies whether perceptually arresting tasks with high levels of visuospatial information have the potential to reduce intrusive memories of traumatic film fragments in the consolidation phase immediately following film viewing. Specifically, we investigated whether simple rating tasks with different degrees of visuospatial complexity (Study 1) or active engagement with complex visual stimuli via imagery (Study 2) would reduce the formation of intrusive memories, as compared with no-task control conditions.

## Study 1

In Study 1, we used traumatic film fragments to induce intrusive memories in healthy individuals and compared the effects of three conditions, differing in the number and complexity of visual items involved in a task given directly after film viewing. In particular, we employed a highly visually arresting (H-VA) task that exposed participants to high levels of visuospatial information (i.e., arresting and attention-grabbing displays), but relatively low working memory load. As control conditions, we used a similar low visually arresting (L-VA) task with little visuospatial information, and a no-task resting condition. We hypothesized that the group performing the H-VA task would report fewer intrusive memories than both the L-VA group and the no-task condition. We also assessed individual differences in WMC using an operation span (OSPAN; [32]) task at baseline, in order to test whether a visuospatial stimulation would be particularly beneficial to individuals with lower WMC. Accordingly, we expected WMC to be a better predictor of fewer intrusive memories in the H-VA condition.

## Materials and methods

### Participants

One-hundred twenty healthy participants (40 per condition; 82.5% women) aged between 18 and 34 years ($M$ = 21.8, $SD$ = 2.9) completed Study 1. On the basis of previous reported effect sizes (e.g., [14] reported $\eta2$ = .083 for memory emotionality comparing visual noise to a control condition], we determined a priori that a sample size of 120 would be adequate to detect an effect size of $f$ = 0.30 in an omnibus ANOVA with 3 groups with ($\eta2p$ = .083, $\alpha$ = .05) with a power $(1 - \beta) > .80$ (actual power = .84). Eligibility for participation was established using a

self-report screening listing the following exclusion criteria: psychological or psychiatric complaints in the past 2 years, severe neurological condition or injury (e.g., epilepsy), pregnancy, current psychoactive medication, alcohol or drug abuse, and having experienced serious traumatic events. Participants were furthermore required to have normal or corrected-to-normal vision and a good understanding of Dutch. All participants gave written informed consent prior to inclusion. Study 1 and Study 2 were approved by the research ethics committee of the Behavioural Science Institute, Radboud University Nijmegen (ECSW2015-1105-310).

## Trauma films and PTSD-analogue symptoms

Participants viewed a compilation of traumatic video fragments lasting about 14 min. The fragments depicted medical surgeries, fatal accidents, and war-related atrocities [33]. Participants were instructed to watch the scenes as if they were a witnessing bystander. In order to enhance immersion, reduce distraction, and create a sense of immobility, participants were provided with headphones and asked to place their head on a chin rest at 60 cm distance from the screen, while being instructed to move as little as possible during film viewing [34]. During film viewing, the experimenter remained in the room in order to monitor the participant's well-being. To avoid memory modulation through social feedback (e.g., [35]), the experimenter was seated outside the participant's field of view and did not engage in discussions about the content or emotionality of the fragments.

To assess intrusive memories related to the film fragments, participants were supplied with a one-week diary ([8]; translated to Dutch), in which they recorded sudden, involuntary memories as soon as they occurred, or the absence of involuntary memories at least twice a day. For each memory, they were also asked to indicate the content and trigger (for verification), whether it was predominantly based on images, thoughts, or both. Finally, they rated the subjective distress caused by each intrusion on an eleven-point scale (range: 0 = *not at all* to 10 = *extremely*). Intrusion frequencies were log-transformed prior to the analyses to correct for their typical right-skewed distribution. Untransformed descriptive statistics are provided in the results section for better readability. Distress scores were averaged across memories, whereby zero was entered if no intrusion had occurred.

We additionally used the Impact of Events Scale (IES; [36]) as a retrospective measure of intrusion-related distress and overall PTSD-analogue symptoms at one-week follow-up. The instructions of the questionnaire were adapted to measure symptoms specifically related to the film fragments. Its Intrusion Symptoms subscale (7 items; e.g., *I had waves of strong feelings about it*; α = .79) was of particular interest, while the total score (including a subscale on avoidance symptoms; α = .83) served as a measure of overall PTSD symptoms.

## Visuospatial rating task

A simple visuospatial rating task was devised to expose participants either to low visually arresting (L-VA) or to high visually arresting (H-VA) displays, depending on their condition, while keeping executive demands to a minimum. For this purpose, participants were instructed to attentively watch a series of images during 15 blocks lasting 30 s. Each block was followed by valence and arousal ratings using Self-Assessment Manikin (SAM; [37]) scales ranging from 1 to 9. Furthermore, participants completed two 100 mm visual analogue scales (VAS) asking '*How impressive or overwhelming did you find these images*?' and '*How fascinating, interesting, or original did you find these images*?' (0 = *not at all*, 100 = *very much*), in order to assess the ability of each clip to attract the viewer's attention. For the analyses, each type of rating were averaged per participant across all stimulation periods.

In the H-VA condition, the series of images consisted of 15 video fragments depicting complex abstract moving displays with many visual details (e.g., landscapes built from changing geometrical shapes, viewed from changing perspectives). The fragments were excerpts from 3D fractal animation artworks by Jérémie Brunet ([38]; "Like in a dream II", "Far away", "Weird planet", "Yellow box"). In the L-VA condition, these clips were replaced with 6 still pictures, each shown for 5 s. In order to create a stimulation condition with comparable low-level features (e.g., colours, luminosity, low frequency spatial patterns), the pictures were screenshots of the 3D animations used in the H-VA condition, heavily blurred by means of a 100 px Gaussian filter. In the third, no-task condition, participants were asked to sit quietly for a comparable amount of time.

## Operation span

To assess individual differences in WMC, we used an automated version [32] of the working operation span task (OSPAN; [39]). This task requires participants to remember a sequence of letter stimuli, next to verifying a mathematical equation before seeing each letter. In 4 practice trials, participants initially learned to memorize and recite short letter sequences by selecting the letters in correct order from a 4×3 letter matrix using the computer mouse. Then, they performed 15 math solving trials, in which participants were asked to solve a simple equation, and to verify a solution that was provided on the next screen. Mean reaction time on these trials plus 2.5 *SD* subsequently served as a time limit for math problems in the remainder of the task. In the final practice phase, participants performed 3 trials of letter recall with a set size of 2, whereby each letter was preceded by a math problem. Finally, the real testing phase continued with set sizes increasing from 3 to 7, with 3 sets of each size. An absolute OSPAN score was calculated as the sum of the sizes of all perfectly recalled sets. Data from three participants were removed because they remembered exceptionally few letters (<20) or committed too many math errors (>40). On average, all other participant remembered 60.1 words (*SD* = 10.3) and committed 4.1 math errors (*SD* = 2.6).

## Affective responses

Mood responses to trauma film viewing were monitored using the Dutch Positive and Negative Affect Schedule (PANAS), state version [40], consisting of two 10-item subscales for negative affect (NA; αs > .80) and positive affect (PA; αs > .84). The PANAS ratings were complemented by four 100 mm visual analogue scales (VAS) measuring current levels of specific negative emotions (sad, fearful, shocked, angry).

## Individual differences in involuntary cognition

To quantify baseline individual differences in the frequency of involuntary memories, we administered the Involuntary Autobiographical Memory Inventory in English (IAMI; [41]). The IAMI consists of two 10-item subscales measuring the frequency of involuntary thoughts on 5-point scales (0 = *never*, 4 = *once an hour or more*) separately for future (α = .90) and past events (α = .90; total score α = .94). The IAMI is supplemented by two similar 5-item subscales measuring voluntary thoughts about future (α = .81) and past events (α = .88; total score α = .91).

## Procedure

Participants were invited to two individual laboratory sessions separated by a one-week interval, during which they completed the intrusion diary. In each session, they were seated in front of a computer screen in a sound-isolated testing cubicle. They initially completed the IAMI

and performed the OSPAN task. Next, they viewed the trauma film fragments, preceded and followed by current mood assessments. Directly afterwards, they were randomly assigned to either the H-VA, L-VA, or no-task condition and performed the respective task. Next, they were provided with the diary and given extensive instructions on its use. Upon their return to the laboratory one week later, participants handed in their diary and the experimenter briefly checked all responses for readability and adherence to the instructions. Participants then completed the IES. Finally, exit questions assessed participants' own judgement of how accurate their diary was, as well as demand characteristics. Finally, they were debriefed, compensated, and dismissed.

## Statistical analyses

The main hypotheses were tested by means of ANOVA and multiple regression analyses. In case of violations of the sphericity assumption, we report Greenhouse-Geisser epsilon and corrected $p$-values along with uncorrected degrees of freedom. Alpha was set at 0.05 (two-tailed) for all tests. Standard frequentist models were computed in SPSS (version 25) and complemented with Bayes factors ($BF$) computed in JASP (version 0.9; [42]) using default priors. $BF_{10}$ reflects the likelihood of the alternative (H1) over the null hypothesis (H0) given the data, whereby the prior assumption was that H1 and H0 are equally likely. Main effects were tested against a null with participants as a sole predictor. Interaction effects were tested against models including the respective main or lower order interaction effects. Evidence is conventionally considered "*anecdotal*" with $BF$s > 1, "*moderate*" with $BF$ > 3, "*strong*" with $BF$ > 10, and "*very strong*" with $BF$ > 30. Either $BF_{10}$ or $BF_{01}$ is reported, whichever is greater than 1. To render extreme $BF$s more readable, we provide natural log-transformed values in some analyses. The analysed dataset can be obtained via the Open Science Framework using the following link: https://osf.io/f38cd/.

## Results

### Baseline differences between conditions

Participants in the three experimental conditions did not differ in mean age, $F(2,117) = 0.62$, $p = .54$, $BF_{01} = 10.1$, or sex distribution, $\chi^2(2) = 0.3$, $p = .84$, $BF_{01} = 7.6$. The groups also did not differ in mean OSPAN scores (overall $M = 44.2$; $SD = 15.0$), $F(2,114) = 0.85$, $p = .43$, $BF_{01} = 6.1$, or on IAMI total scores (overall $M = 2.1$; $SD = 0.7$), $F(2,117) = 0.16$, $p = .85$, $BF_{01} = 11.0$.

### Affective responses to the trauma films

A 2 (Time: pre-film, post-film) by 3 (Condition) mixed ANOVA revealed a main effect of Time for NA, $F(1,117) = 96.47$, $p < .001$, $\eta^2p = .45$, $\log(BF_{10}) = 32.5$ with scores increasing in response to film viewing from 12.5 ($SE = 0.3$) to 17.4 points ($SE = 0.6$). Similarly, PA decreased with time, $F(1,117) = 78.78$, $p < .001$, $\eta^2p = .40$, $\log(BF_{10}) = 27.3$, from 27.5 ($SE = 0.5$) to 23.1 points ($SE = 0.6$). Also scores on each of the four VAS scores increased with time, all $F$s > 37.94, $p$s < .001, $\eta^2p$s > .24, $\log(BF_{10}) > 13.7$ (anxious: $M_{difference} = 13.2$, $SD = 23.5$; shocked: $M_{difference} = 36.8$, $SD = 29.6$; angry: $M_{difference} = 13.0$, $SD = 21.6$; sad: $M_{difference} = 20.7$, $SD = 24.4$). For none of the affective responses were there any effects involving condition, $F$s < 1.67, $p > .19$, $\eta^2p < .03$, $BF$s$_{01}$ > 2.0.

### Visuospatial rating task

Table 1 summarizes the mean subjective ratings from the visual rating task, separately for the H-VA and L-VA conditions (note that these data were not collected in the resting condition).

**Table 1. Mean ratings of the visual displays in the H-VA and L-VA condition.**

| | Condition | | | | $t$ ($df$ = 78) | $p$ | $d$ | $BF_{10}$ |
|---|---|---|---|---|---|---|---|---|
| | H-VA | | L-VA | | | | | |
| | M, SD | 95% CI | M, SD | 95% CI | | | | |
| Impressive (0–100) | 44.0, 17.3 | 38.5, 49.5 | 25.0, 19.8 | 18.7, 31.4 | 4.57 | < .001 | 1.02 | 1019.70 |
| Interesting (0–100) | 47.8, 14.1 | 43.3, 52.3 | 30.3, 21.1 | 23.5, 37.0 | 4.37 | < .001 | 0.98 | 519.00 |
| Valence (1–9) | 4.6, 1.0 | 4.3, 5.0 | 4.8, 0.9 | 4.5, 5.1 | -0.78 | .437 | 0.21 | 0.30 |
| Arousal (1–9) | 5.6, 1.2 | 5.2, 6.0 | 6.3, 1.4 | 5.8, 6.7 | -2.28 | .026 | 0.54 | 2.12 |

*Note.* H-VA = high visually arresting condition; L-VA = low visually arresting condition.

As expected, participants in the H-VA condition rated the visual displays as visually more impressive and more interesting, compared to participants in the L-VA condition. Furthermore, there were no differences between conditions in the valence ratings. Participants in the L-VA condition tended to rate the displays as slightly more arousing, although the corresponding BF is small, providing only "anecdotal" evidence (see Table 1). Notably, the H-VA group rated the stimuli not only as more impressive and interesting on average, but also used a wider range of scores for the individual stimuli than the L-VA group, both for impressiveness (Range $M_{Difference}$ = 20.8; SE = 5.4), $t$(78) = 3.83, $p$ < .001, $d$ = 0.86, $BF_{10}$ = 95.89, and for interest (Range $M_{Difference}$ = 17.0; SE = 4.9), $t$(78) = 3.83, $p$ < .001, $d$ = 0.78, $BF_{10}$ = 35.97. Consequently, when examining the maximum rather than averages scores across stimuli, the mean differences between the conditions become even more pronounced for both impressiveness (H-VA: $M$ = 74.4; SD = 23.1; L-VA: $M$ = 48.0 SD = 28.4), $t$(78) = 4.55, $p$ < .001, $d$ = 1.02, $BF_{10}$ = 934.92, and interest (H-VA: $M$ = 77.6; SD = 15.0; L-VA: $M$ = 57.6 SD = 28.7), $t$(78) = 3.89, $p$ < .001, $d$ = 0.87, $BF_{10}$ = 117.66.

## PTSD analogue symptoms

Means, standard deviations, as well as comparison statistics for intrusive memories and related symptoms are summarized in Table 2. As can be seen, our main analyses on weekly intrusion numbers indicated no differences between the three groups, and there also were no differences in other intrusion-related symptoms. Instead, we found moderate to strong support for the null hypotheses.

In addition to these outcomes, we explored potential group differences in intrusion development over the one-week period by subjecting log-transformed intrusive memories to a 7

**Table 2. PTSD analogue symptoms per experimental condition in Study 1.**

| | Condition | | | | | | $F$ (2,117) | $p$ | $\eta^2 p$ | $BF_{01}$ |
|---|---|---|---|---|---|---|---|---|---|---|
| | H-VA | | L-VA | | No-task | | | | | |
| | M, SD | 95% CI | M, SD | 95% CI | M, SD | 95% CI | | | | |
| Intrusions (all) | 6.0, 4.2 | 4.7, 7.3 | 5.6, 4.0 | 4.3, 6.9 | 5.4, 4.0 | 4.1, 6.7 | 0.28 | .76 | >.01 | 10.0 |
| Image | 5.1, 3.6 | 3.9, 6.2 | 5.0, 4.0 | 3.7, 6.2 | 3.9, 2.6 | 3.1, 4.8 | 0.83 | .44 | .01 | 6.3 |
| Thought | 2.3, 2.8 | 1.4, 3.2 | 2.1, 2.9 | 1.2, 3.1 | 2.5, 3.2 | 1.5, 3.6 | 0.20 | .82 | >.01 | 10.7 |
| Distress (0–10) | 3.9, 2.1 | 3.3, 4.5 | 3.1, 1.8 | 2.6, 3.7 | 3.3, 1.8 | 2.7, 3.9 | 1.78 | .17 | .03 | 2.9 |
| IES intrusions | 9.0, 5.6 | 7.3, 10.6 | 7.4, 5.0 | 5.7, 9.0 | 8.1, 5.3 | 6.4, 9.7 | 0.92 | .40 | .02 | 5.9 |
| IES total | 14.5, 10.0 | 11.4, 17.5 | 13.2, 10.3 | 10.1, 16.2 | 12.7, 8.8 | 9.6, 15.7 | 0.36 | .70 | >.01 | 9.4 |

*Note.* H-VA = high visually arresting condition; L-VA = low visually arresting condition. IES = Impact of Event Scale.

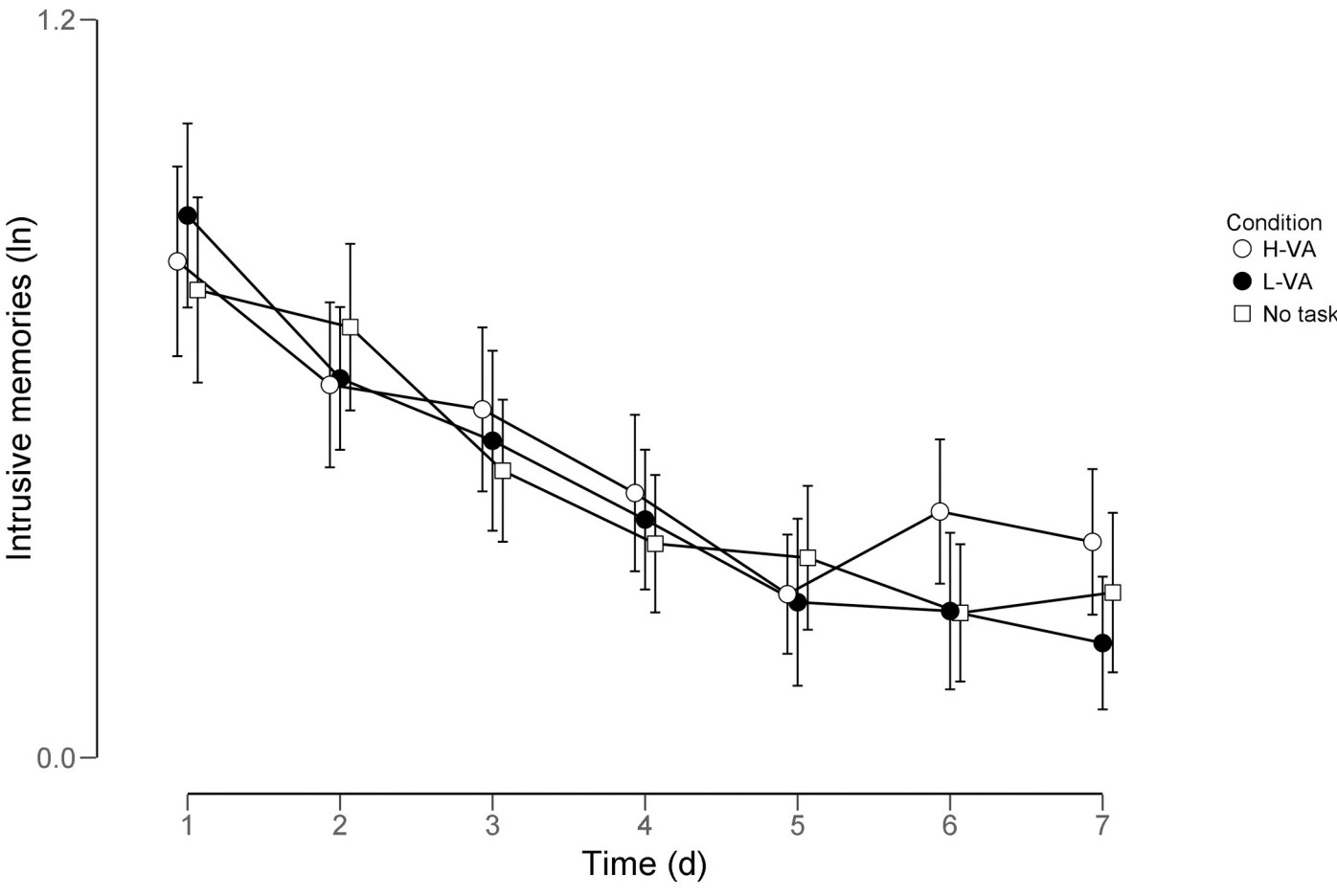

**Fig 1. Intrusive memories (log-transformed) per condition per day.** Error bars indicate 95% confidence intervals.

(Time: 7 days) by 3 (Condition) mixed ANOVA (see Fig 1). There only was a Time main effect, $F(6,702) = 33.57$, $p < .001$, $\eta^2 p = .22$, $\log(BF_{10}) = 75.2$. Meanwhile, the data strongly favoured the absence of a main effect for Condition, $F(2,117) = 0.39$, $p = .68$, $\eta^2 p = .01$, $BF_{01} = 19.3$, as well as an absence of a Condition by Time interaction, $F(12,702) = 0.94$, $p = .50$, $\eta^2 p = .01$, $BF_{01} = 174.2$. When entering IAMI total scores and Negative Affect increase as additional covariates (i.e., factors potentially accounting for individual differences in intrusions), additional main effects emerged only for IAMI total scores, $F(1,115) = 12.44$, $p = .001$, $\eta^2 p = .10$, $BF_{10} = 200.8$, and NA increase, $F(1,115) = 5.94$, $p = .016$, $\eta^2 p = .05$, $BF_{10} = 11.8$.

## OSPAN

To test for linear associations between OSPAN performance and intrusive memories and potential moderation effects of our experimental manipulation, we performed hierarchical linear regression analyses with three steps. First, OSPAN scores were entered as predictors of intrusion symptoms, followed by two dummy variables representing the L-VA and H-VA groups, respectively (i.e., no-task served as the reference group). In the third and final step, we entered interaction terms for L-VA and H-VA (i.e., $z$-transformed OSPAN scores × group dummy).

The first step did not yield any model explaining a significant proportion of variance, $r^2s < .02$, $Fs(1,115) < 2.2$, $ps > .14$, $BF_{01}$ of coefficients ranging between 1.90 and 5.08. In line with

the group analyses in the previous section, the group dummy variables did not add any explained variance either, $r^2$ changes < .032, $Fs$ (2,113) < 1.9, $ps$ > .16, $BF_{01}$ range: 1.00–8.45. Finally, adding the two group by OSPAN score interaction terms also failed to explain additional variance, $r^2$ changes < .046, $Fs$ (2,111) < 1.5, $ps$ > .23, with $BFs_{01}$ 1.22 and 3.39 in favour of the null hypotheses and largely in the "anecdotal" range. We additionally performed similar regression models to directly compare the H-VA and the L-VA groups (H-VA serving as the reference group). These analyses did not reveal any additional effects.

### Demand effects and diary accuracy

On average, participants in all three conditions did not assume that performing a visuospatial task following viewing of the trauma film had (or would have had) a notable influence on intrusive memories (anchors: -10 = *extreme decrease*; 10 = *extreme increase*), $M$ = -0.4, $SD$ = 3.6, with no differences between conditions, $F$ (2,117) = 1.42, $p$ = .25, $\eta^2$p = .02, $BF_{01}$ = 3.9. Moreover, participants generally indicated that their diary was fairly accurate (anchors: 0 = *not accurate at all*; 10 = *very accurate*), $M$ = 8.4, $SD$ = 1.4, again, with no differences between conditions, $F$ (2,117) = 1.50, $p$ = .23, $\eta^2$p = .03, $BF_{01}$ = 3.7. When omitting 8 participants with accuracy ratings below 7 from the analyses, all conclusions drawn from the group analyses reported above remain virtually unchanged.

### Summary and discussion

Overall, Study 1 provides moderate to strong evidence that performing a simple visual rating task directly after viewing a trauma film has no beneficial effect on the development of intrusive memories, as compared to a no-task control condition. Furthermore, it was irrelevant whether participants viewed highly arresting, impressive, and interesting images, as opposed to low visually and less interesting arresting displays. Our data also provide anecdotal to moderate evidence against a direct or moderated association of OSPAN scores with intrusive memories. Taken together, our findings seem to suggest that the build-up of intrusive memories is not directly dependent on WMC, while engaging with visuospatially arresting images in a relatively passive rating task does not mimic the beneficial effects that have been reported using more active visuospatial tasks, such as pattern tapping and the computer game Tetris [4–6].

### Study 2

Based on the findings reported above, Study 2 addressed two cognitive components that may be required for a reduction of intrusive memories, in addition to the ability of the H-VA task to capture attention and interest; (a) high working memory load and (b) concurrent activation of the aversive memories. That is, a working memory account predicts that interference with intrusive memories is a function of load on working memory resources, in particular on visual short-term memory (VSTM) resources [8, 14, 19]. Meanwhile, some studies suggest that the modulation of intrusive memories requires active competition for working memory resources between trauma film memories and the concurrent task. For instance, an eye-movement intervention was found to effectively reduce the emotionality and vividness of aversive memories, but only when these memories were actively held in mind [25, 43]. Similarly, once the memories of a trauma film have been consolidated (i.e., after a longer interval), playing the video game Tetris has been found to reduce intrusive memories of trauma film only when memories of the film had been reactivated prior to gameplay [44, 45]. Although reactivation may not be required in the period following film viewing (e.g., [7]), many studies have still included a brief reminder task (e.g., [9]).

To test these possibilities, we devised an *imagery* condition, in which we instructed participants to memorize and to actively imagine the H-VA displays from Study 1 in order to maximize working memory load. We expected that performing this task after trauma film viewing would reduce intrusive memories compared to a no-task comparison group. Next, in order to test the role of concurrent activation of trauma film memories, a second experimental group performed the same task, while being confronted repeatedly with trauma-film reminder pictures (i.e., *imagery + reminder* condition). Notably, it is conceivable that this additional rehearsal of reminder pictures would lead to more intrusive memories, potentially cancelling out beneficial effects of working memory engagement. However, the degree to which participants would engage in imagery should be negatively associated with intrusive memories. Therefore, we included a recognition test for the H-VA displays at the end of the experiment as a proxy to active working memory engagement during the task. We expected better recognition performance to correlate with fewer intrusive memories, and particularly so in the imagery + reminder condition.

## Participants

One-hundred fourteen participants (78.9% women) aged between 18 and 31 years ($M$ = 21.4, $SD$ = 2.8) completed this study. Similar to Study 1, the sample size was determined to be adequate to detect an effect size of $f$ = 0.30 in an omnibus ANOVA with 3 groups with with a power $(1 − β) > .80$ (actual power = .81). Eligibility and recruitment were kept in line with Study 1 with one exception; due to an increase in international students among the student population, we decided to drop fluency in Dutch as an inclusion criterion and conducted the entire study in English.

## Imagery task

In the imagery condition, participants were alternately asked to watch high visually arresting displays and then to actively remember and imagine them. In particular, they were shown a total of 7 clips lasting 30 s with the instruction to watch them attentively while trying to memorize as many details as possible. As in Study 1, each clip was followed by valence, arousal, impressiveness and interest ratings. After a short break of 5 s, participants performed an imagery trial lasting 30 s. Here, they were asked to vividly imagine the previously seen clip in their mind's eye in as much detail as possible. They could initiate the trial in a self-paced manner and were free to keep their eyes closed or open. At the beginning of the imagery trial, a 1-s fragment of the H-VA clip was shown as a memory aid, after which a white screen was displayed. As another memory aid, participants were allowed to replay up to five additional 1-s fragments during the trial at any time by using the space button. The end of the imagery trial was signalled by two tones. Finally, an additional VAS rating was used to measure participants' subjective vividness of their imagery.

The 30 s clips were taken from the same selection of H-VA clips as in Study 1. In order to devise a recognition test for the film clips, we selected 7 pairs of clips to create two comparable sets consisting of 7 clips each. Each clip pair consisted of two non-overlapping sequences from the same source film. This enabled us to use one set of clips as old and the other as new probes in the recognition test (see below). In order to facilitate distinguishing each pair of clips from the same source film, the clips were mildly video-edited with opposite colour temperature adjustments.

## Recognition test

To quantify recognition memory, we presented participants with nine different 2 s fragments of each of the 14 video clips. On each trial, they were asked to indicate as quickly and as

accurately as possible whether the fragment belonged to a clip that they had seen before (old) or not (new) by using a left or right response button on the keyboard. Trial order and response key assignment was fully randomized for each participant with the restriction that no more than 4 trials in a row depicted the same clip or would require the same correct response. The extracted response data are summarized in terms of hit and false alarm rates. We then calculated a discrimination index d' and criterion C according to Signal Detection Theory [46].

## Procedure

Similar to Study 1, participants were invited to two individual laboratory sessions separated by a one-week interval. The materials for the trauma film paradigm and all questionnaire-based assessments were the same as in Study 1, using the corresponding English language versions or translations of the materials. The crucial difference was that participants performed either the imagery task with or without trauma film reminder pictures or were assigned to the no-task control condition.

## Statistical analyses

The hypotheses were addressed with the same statistical methods as in Study 1. In the analyses of experimental between-group effects, we additionally considered the version of the visuospatial rating task as an additional factor. The analysed dataset can be obtained via the Open Science Framework using the following link: https://osf.io/f38cd/.

## Results

### Baseline differences

Participants in the three experimental conditions did not differ in mean age, $F(2,111) = 1.21$, $p = .30$, $BF_{01} = 5.0$, or sex distribution, $\chi^2(2) = 3.1$, $p = .21$, $BF_{01} = 4.9$. The groups also did not differ in IAMI total scores, $F(2,111) = 0.54$, $p = .58$, $BF_{01} = 7.8$.

### Affective responses

A 2 (Time: pre-film, post-film) by 3 (Condition) mixed ANOVA revealed a main effect of Time for NA, $F(1,111) = 143.60$, $p < .001$, $\eta^2p = .56$, $\log(BF_{10}) = 47.0$ with scores increasing in response to film viewing from 14.5 ($SE = 0.4$) to 23.1 points ($SE = 0.7$). Similarly, PA decreased from 29.4 ($SE = 0.6$) to 25.1 points ($SE = 0.6$), $F(1,111) = 50.70$, $p < .001$, $\eta^2p = .31$, $\log(BF_{10}) = 18.0$, while scores on the four VAS increased, all $Fs > 72.96$, $ps < .001$, $\eta^2ps > .39$, $\log(BF_{10}) > 28.9$ (anxious: $M_{difference} = 24.5$, $SD = 27.8$; shocked: $M_{difference} = 51.0$, $SD = 29.8$; angry: $M_{difference} = 22.7$, $SD = 28.2$; sad: $M_{difference} = 32.8$, $SD = 30.2$). In addition, there was an unintended Condition main effect for PA, $F(2,111) = 5.46$, $p = .001$, $\eta^2p = .09$, $BF_{10} = 7.4$, and a Time by Condition interaction for feeling anxious, $F(2,111) = 4.49$, $p = .013$, $\eta^2p = .08$, $BF_{10} = 3.2$. These effects were due to relatively higher overall PA scores and a lower increase in anxiety for the no-task condition, compared to the other two groups. Similar main or interaction effects could not be ascertained for any other item, all $ps > .055$, $BFs_{01} > 1.1$.

### Ratings task

We found no evidence for or against differences in the ratings in the visuospatial imagery task between the two experimental groups, all $Fs(1,74) < 1.18$, $ps > .27$, $\eta^2ps < .02$, $BFs_{01} > 1.7$. When video clip set was entered as an additional between-subjects factor (unbalanced due to random allocation; $ns = 34$ and $42$), there was no evidence for or against additional main or interaction effects, $Fs(1,72) < 2.6$, $ps > .11$, $\eta^2ps < .03$, $BFs_{01} > 1.2$. Comparably to the H-VA

**Table 3. PTSD analogue symptoms per experimental condition in Study 2.**

| | Condition | | | | | | F (2,111) | p | BF_01 |
|---|---|---|---|---|---|---|---|---|---|
| | Imagery (n = 39) | | Imagery + Reminder (n = 38) | | No-task (n = 37) | | | | |
| | M, SD | 95% CI | M, SD | 95% CI | M, SD | 95% CI | | | |
| Intrusions (all) | 6.7, 5.0 | 5.1, 8.3 | 8.4, 6.4 | 6.3, 10.5 | 5.3, 3.9 | 4.0, 6.6 | 2.34 | .101 | 1.8 |
| Image | 5.4, 4.6 | 3.9, 6.9 | 7.0, 6.2 | 5.0, 9.0 | 4.3, 3.5 | 3.1, 5.5 | 2.37 | .099 | 1.8 |
| Thought | 3.4, 3.4 | 2.3, 4.5 | 3.4, 4.0 | 2.1, 4.8 | 2.4, 2.7 | 1.5, 3.3 | 0.77 | .466 | 6.5 |
| Distress (0–10) | 3.6, 1.8 | 2.9, 4.2 | 3.8, 1.9 | 3.1, 4.4 | 3.1, 2.4 | 2.2, 3.5 | 2.19 | .116 | 2.0 |
| IES–intrusions | 9.4, 6.8 | 7.4, 11.3 | 10.8, 6.1 | 8.8, 12.8 | 8.2, 5.7 | 6.2, 10.2 | 1.58 | .210 | 3.3 |
| IES–total | 16.3, 12.2 | 12.7, 19.9 | 19.1, 10.2 | 15.4, 22.8 | 17.0, 11.2 | 13.3, 20.6 | 0.64 | .530 | 7.1 |

*Note*. IES = Impact of Event Scale.

group in Study 1, participants rated the displays as fairly neutral ($M = 4.1$, $SD = 1.1$), mildly arousing ($M = 6.0$, $SD = 1.3$), and moderately impressive ($M = 51.5$, $SD = 16.1$), with slightly higher interest ratings ($M = 57.1$, $SD = 17.1$). Participants rated the subjective vividness during the imagery trials with an average of 56.8 ($SD = 14.9$).

## PTSD analogue symptoms

Means, standard deviations, as well as comparison statistics for intrusive memories and related symptoms are summarized in Table 3. As can be seen, the findings tend to support the null hypotheses, although the evidence is mostly in the "anecdotal" range. However, most mean scores in the no-task condition were descriptively lower than the other two conditions. Therefore, unsurprisingly, one-sided Bayesian $t$-tests more consistently favour $H_0$ over the hypothesis that the imagery condition would reduce intrusive memories compared to the no-task, all $BFs_{01} > 3.4$, or that the imagery + reminder condition would, all $BFs_{01} > 7.1$.

An exploratory 7 (Time: 7 days) by 3 (Condition) mixed ANOVA further corroborated these findings (see Fig 2). Next to a main effect for Time, $F (6,666) = 40.13$, $\varepsilon = .84$, $p < .001$, $\eta^2p = .27$, $\log(BF_{10}) = 89.8$, the evidence for a Condition main effect was inconclusive, $F (2,111) = 2.75$, $p = .068$, $\eta^2p = .05$, $BF_{01} = 2.1$, while an interaction can virtually be ruled out,

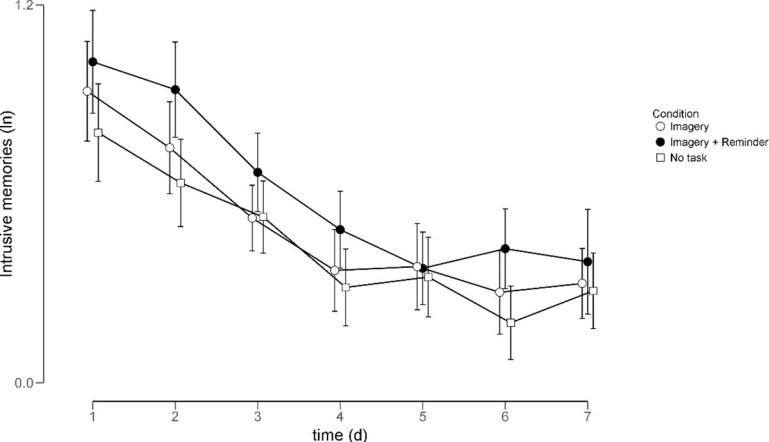

**Fig 2. Intrusive memories (log-transformed) per condition per day.** Error bars indicate 95% confidence intervals.

**Table 4. Recognition memory performance in the two imagery conditions.**

| Recognition memory | Condition | | | | Condition main effect | | | |
|---|---|---|---|---|---|---|---|---|
| | Imagery | | Imagery + Reminder | | | | | |
| | Clip set A | Clip set B | Clip set A | Clip set B | $F\ (1,72)$ | $p$ | $\eta^2 p$ | $BF_{01}$ |
| Hit rate (%) | 60.3 (11.3) | 69.5 (13.2) | 63.9 (13.1) | 71.6 (13.6) | 0.81 | .373 | .01 | 4.1 |
| FA rate (%) | 25.6 (11.9) | 21.8 (11.2) | 27.8 (13.2) | 24.7 (11.6) | 0.80 | .375 | .01 | 2.2 |
| d' | 0.98 (0.14) | 1.40 (0.62) | 1.05 (0.73) | 1.37 (0.57) | 0.02 | .877 | < .01 | 3.8 |
| C | 0.21 (0.25) | 0.14 (0.27) | 0.14 (0.27) | 0.05 (0.33) | 1.33 | .252 | .02 | 2.9 |

*Note.* Values in brackets denote standard deviations.

$F\ (12,666) = 0.85$, $p = .846$, $\eta^2 p = .01$, $BF_{01} = 626.8$. This pattern of findings did not change when IAMI total and Negative Affect increase were entered as additional covariates.

### Recognition memory and intrusive memories

Table 4 summarizes mean scores of recognition memory performance, separately for the two different test versions. As can be seen in Table 4, no condition main effects emerged. The analyses only revealed minor differences between the two test versions in hit rates and discrimination scores, $Fs\ (1,72) > 6.21$, $ps < .015$, $\eta^2 ps > .08$, $BFs_{10} > 3.8$. Similar effects were not evident for FA rates or C, $Fs < 1.6$, $ps > .221$, $\eta^2 ps < .03$, $BFs_{01} > 1.6$. There was no evidence to suggest Test version by Condition interactions, $Fs\ (1,72) < 0.2$, $ps > .73$, $\eta^2 ps < .01$, $BFs_{01} > 2.8$.

To test for linear associations between recognition memory performance and intrusive memories and for moderation by task (imagery vs. imagery + reminder), we performed hierarchical linear regression analyses with three steps. We first entered d' scores, followed by condition and test version (dummy variables), and, in the final step, the interaction term for recognition performance by condition. The first step revealed explained variance for mean intrusion distress, $r^2 = .077$, $F\ (1,74) = 6.17$, $p = .015$, $BF_{10} = 3.2$, but not for any other intrusion symptom, $r^2 s < .012$, $Fs\ (1,74) < 0.95$, $ps > .34$, $BF_{01}$ range: 2.8–4.2. In particular, higher discrimination performance predicted lower distress levels ($\beta = -.277$). Extending the group analyses described above, neither the second nor third regression step explained additional variance, $r^2$ changes $< .043$, $Fs < 1.45$, $ps > .243$, $BF_{01}$ range: 1.7–8.0.

### Demand effects and diary accuracy

As in study 1, participants generally assumed no influence of performing a visuospatial task on intrusive memories, $M = 0.2$ $SD = 4.2$, with no differences between conditions, $F\ (2,110) = 0.12$, $p = .885$, $\eta^2 p < .01$, $BF_{01} = 10.9$. Participants generally indicated that their diary was fairly accurate, $M = 7.8$, $SD = 1.7$, although slight differences between conditions could not be ruled out, $F\ (2,110) = 3.34$, $p = .039$, $\eta^2 p = .06$, $BF_{10} = 1.3$. However, when 19 participants with accuracy ratings below 7 were removed from the analyses, the conclusions drawn from the group analyses reported above remain virtually unchanged.

## Discussion

In the present paper, we examined whether arresting, attention-grabbing, and interesting visual stimuli can be used to reduce the development of intrusive memories in the consolidation phase immediately following exposure to a trauma film. Furthermore, we explored whether beneficial effects on intrusive memories would depend on working memory resources

or the concurrent activation of trauma film memories. Against our expectations, our data provide moderate to strong evidence that performing a visuospatial rating task after trauma-film viewing has no beneficial effects compared to no task, irrespective of the visuospatial complexity of the stimuli and of working memory demands. Moreover, we found anecdotal to moderate evidence contradicting a direct link between WMC and intrusive memories. That is, OSPAN scores were neither directly associated with intrusive memories in Study 1, nor did they moderate condition effects. Study 2 mirrored and extended these null findings, as the degree of engagement in the visually arresting task–measured in terms of recognition memory after one week–was statistically unrelated to intrusion development. A single exception to this pattern in our data was an association between higher task engagement and lower intrusion distress.

Our findings appear to be at odds with a growing body of studies showing that engaging in various types of tasks shortly after exposure to traumatic films can reduce the development of intrusive memories [7–10]. A popular account for these effects is that traumatic memories are sensitive to disruption by a dual task while they undergo the processes of consolidation or reconsolidation/re-encoding (i.e., during and shortly after formation or activation of the memory trace). Moreover, a widely held view is that high levels of working memory engagement constitute the principal cognitive mechanism by which dual tasks reduce the build-up of intrusive memories (e.g., [8, 14, 19]). Indeed, this might explain why beneficial effects have been documented for very different types of cognitive tasks, including the video game Tetris, pattern tapping, or N-back tasks (e.g., [14, 16, 19]), although not all findings in the literature are consistent with this interpretation (e.g., [29, 30]). As we discuss in the following, the present findings demonstrate that these assumptions lack one or more boundary conditions that need to be satisfied in order to effectively reduce intrusive memories.

## Attention, task demands, and working memory capacity

To begin with, our results indicate that engagement with visuospatial stimuli alone after a film is insufficient to interfere with the development of intrusive memories. That is, both visual stimulation conditions in Study 1 did not differ from one another or from the no-task condition in terms of intrusion development. This suggests that trauma-film memories cannot be modulated by the mere attraction or redirection of attention (e.g., away from distressing imagery) in the consolidation phase following film viewing. This was the case even though we could demonstrate that our H-VA stimuli were clearly superior to those in the L-VA condition in evoking subjective impressiveness and interest. Notably, while all H-VA stimuli were rated as moderately impressive and interesting on average (see Table 1), our analyses revealed that at least some of the H-VA stimuli were rated as highly impressive and interesting by most participants. In addition, our data show that the H-VA clips were equivalent to the L-VA stimuli in valence, and potentially even less arousing. Since both conditions displayed similar levels of intrusive memories, our results suggest that the visuospatial complexity of the stimuli, or the degree to which they are perceived as "arresting", may be irrelevant to the consolidation of trauma-film memories.

To some degree, our results seem to align with the idea that tapping into working memory resources is necessary for a dual task to interfere with the consolidation of traumatic memories (e.g., [8, 14, 19, 20]). However, this view cannot accommodate our finding in Study 2, where we devised a variant of our rating task that was cognitively much more taxing than the one used in Study 1, while continuing to use complex and visually arresting stimuli. Since participants had to memorize and imagine the complex images, this intervention was designed to tap into working memory resources that are especially relevant for visual imagery (e.g., the

visuospatial sketchpad; [18]). Although it has been argued that such a modality-specific intervention should be especially effective, we again found that none of our active conditions differed from the no-task condition in terms of intrusion development. In addition, we found no correlations between recognition memory performance for the complex visual displays–a proxy for task engagement–and intrusive memories. Together, our results suggest that working memory engagement and simultaneous engagement with visually arresting material is not a sufficient condition for a dual task to reduce intrusive memories.

Some recent studies with different visuospatial tasks appear to point in a similar direction. For instance, Asselbergs et al. [29] used two different visuospatial games involving spatial planning and decision making (i.e., controlling an airplane in a 2D capture-and-avoid game; a spatial sorting game) shortly after exposing participants to traumatic films and failed to find any beneficial effects on intrusive memories. Similarly, van Schie et al. [30] had participants perform eye movements or a counting task whilst remembering the hotspot of a traumatic film. In only one out of three experiments was there a reduction in intrusive memories compared to a no-task control condition, and this effect was irrespective of modality (i.e. eye movements and counting). More problematic for the working memory account, both aforementioned series of studies [29, 30] established that their interventions effectively taxed WM resources, demonstrating robust interference effects in a simple dual reaction time task. Van Schie et al. also quantified the degree of dual task interference for each participant but failed to observe a correlation with a reduced intrusion vividness or emotionality, similar to our finding in Study 2 that recognition memory of the complex stimuli was unrelated to intrusive memories. Taken together, the available evidence contravenes the ideas that visuospatial engagement, WM taxation, or a combination of both provide a reliable recipe for the reduction of intrusive memories. On a related note, our data suggest that intrusive memories have no direct relationship with individual differences in WMC–i.e. the amount of goal-relevant information that one is able to hold active in working memory in the presence of distractors. Also, WMC did not statistically modulate any effects of the H-VA or L-VA conditions in Study 1. Thereby, our study adds to a rather mixed evidence in the literature. On the one hand, a number of studies have identified WMC as a direct [22, 23, 47, 48] or indirect [25] factor modulating intrusive memories or thought suppression. On the other hand, several studies failed to replicate or extend these findings [26, 30, 49–51]. Jointly, these findings indicate that the relationship between WMC and intrusive trauma memories may be more subtle than often assumed or may be moderated by factors that are not yet fully established.

In addition to dual-task demands and individual differences in WMC, it has been proposed that the targeted memories (i.e. of the trauma film) need to be activated and compete for imagery resources during the dual task in order to disrupt intrusive memories [25, 43]. This has the effect of making the design similar to studies where the dual task is actually contemporaneous with seeing the film [52, 53]. Indeed, for studies using the video game Tetris to reduce trauma-film intrusions, there is emerging evidence that this task may be especially effective if the target memories had been re-activated prior to gameplay (e.g., [44]), while it may be ineffective in the absence of a reminder task [45]. Arguably, there probably was residual activation of trauma-film memories during the interventions in both of our studies, as we administered the tasks immediately following film viewing. We thus targeted a time window in which the trauma-film contents were still held in short-term memory and underwent memory consolidation [54]. Therefore, our null results in Study 1 appear to contradict Kavanagh et al. [14], who found that simply viewing a screen with flickering squares (labelled "visual noise" by the authors) during recall of emotional autobiographical memories decreased the memories' vividness and emotionality. Similarly, the cognitively more demanding imagery condition in Study 2 would have been expected to produce effects comparable to Tetris gameplay in previous

studies. Critically, even our imagery + reminder condition produced no beneficial effects compared to imagery alone or no task, despite the fact that it maximized the assumed competition between trauma film memories and current task demands.

## Future directions

The above-mentioned considerations indicate the need for careful systematic investigation into the boundary conditions for dual tasks introduced after the traumatic event that aim to reduce intrusive memories. A number of avenues may be particularly promising for future research to follow up. First, the ability of certain cognitive tasks to interfere with the development of intrusive memories may depend on characteristics other than their working memory demands. For example, potential differences between our visuospatial imagery task and pattern tapping or Tetris are that the latter tasks require spatial planning, updating, coordination of motor movements that have to be integrated with working memory, or mapping motor movements onto spatial representations. Following up on our studies, future research could address these factors directly, e.g., by comparing actual Tetris gameplay to passive Tetris-watching.

In addition, the videogame Tetris typically matches the task difficulty to the player's current skill level. Optimal skill-demands compatibility in challenging tasks is known to induce a so-called *flow experience* [55]. Flow experience is characterized by a sense of control, effortless attention, and reduced self-consciousness, and may be critical for the psychobiological impact of gameplay. For instance, it is accompanied by pleasurable feelings and moderately elevated physical arousal and stress hormonal levels (e.g., [56, 57]). In addition, high levels of flow experience may result in altered experiences even after gameplay, including continued replay, as well as intrusive visual images of the game [58]. Speculatively, the beneficial effects that have been reported for Tetris may thus depend on competition for visuospatial resources after, rather than during gameplay. Therefore, an exciting avenue for future studies would be to explore if a task's ability to induce a flow experience, continued replay, and/or intrusive imagery is required in order to interfere with intrusive memories of a trauma film.

These considerations aside, our findings could suggest that the film memories were robust against interference and degradation, even though our intervention targeted the immediate consolidation phase [54]. A reason may have been that there was no need to update and/or correct the memories during this period. Indeed, for memories that have already been consolidated, the presence of a prediction error has been proposed to be necessary in order for a reactivated memory to become labile and susceptible to interference during reconsolidation [59]. For instance, Gotthard and Gura [60] have recently shown that free recall of a positive emotional film could be degraded by having participants perform a visuospatial word search task—but only when participants had received new information about the film. Future studies into intrusive trauma memories may thus want to address the question if degrading intrusive memories requires the presence of novel, conflicting, or surprising information about the traumatic experience, even in the immediate consolidation phase following the traumatic experience.

The present findings also underline that the precise relationship between WMC and intrusive memories merits careful and systematic investigation. A fruitful approach might be to investigate this relationship experimentally by means of WMC trainings. For instance, Bomyea and Amir [47] used two training conditions that required high versus low interference control (see also [49]) and found that participants displayed relatively fewer intrusive thoughts in the laboratory following the high interference control training. However, this effect emerged only after participants had been prompted to actively suppress their intrusions, whereas it was not

present before. Speculatively, this may suggest that the role of WMC in intrusive memories is limited to situations where the affected individual actively tries to suppress the recollections. Notably, the motivation to suppress memories of a trauma film may be relatively low compared to aversive memories in trauma survivors. Accordingly, naturally occurring intrusions in the trauma-film paradigm might be relatively unaffected by WMC, in contrast to more effortful attempts to suppress and forget certain memories. Future research following up on these ideas might benefit from carefully defining and disentangling different types of voluntary and involuntary cognition, because multiple memory traces might be differentially affected by WMC and by interfering visuospatial tasks (e.g., [9, 61]).

In addition to these considerations, future research should bear in mind a few limitations of the present studies. Critically, due to the complex nature of our visuospatial stimuli, there may have been differences or overlap between our experimental conditions on critical characteristics that went undetected. For instance, we used one-week recognition memory performance to assess and control for engagement in the imagery task in Study 2. However, potential engagement effects may have been overshadowed by individual differences in visual memory performance. Similarly, as discussed above, we may have missed potentially important task characteristics such as a flow experience. Finally, we used an analogue sample of healthy participants, so that our results may not translate to traumatized populations. Therefore, complementary studies with trauma survivors are essential before conclusions about practical implications can be drawn.

To conclude, the present studies indicate that interference with the development of intrusive memories requires more than having individuals engage with arresting, attention-grabbing, and interesting visual stimuli. In addition, our data provide some evidence against a direct role for individual differences in WMC or for working memory demands in intrusive memories. This indicates that the conditions under which intrusive memories can be reduced remain to be established in future research.

## Author Contributions

**Conceptualization:** Thomas Meyer, Chris R. Brewin, John A. King, Eni S. Becker.

**Formal analysis:** Thomas Meyer, Marcella L. Woud.

**Investigation:** Thomas Meyer, Desiree Nijmeijer.

**Supervision:** Eni S. Becker.

**Writing – original draft:** Thomas Meyer.

**Writing – review & editing:** Chris R. Brewin, John A. King, Desiree Nijmeijer, Marcella L. Woud, Eni S. Becker.

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
