## [Decision Letter · Decision Letter 0]

24 Sep 2019

PONE-D-19-21303

Arresting visuospatial stimulation is insufficient to disrupt analogue traumatic
intrusions

PLOS ONE

Dear
Dr Thomas
Meyer,

Thank you for submitting your manuscript to PLOS ONE. After careful consideration, we
feel that it has merit but does not fully meet PLOS ONE’s publication criteria as it
currently stands. Therefore, we invite you to submit a revised version of the
manuscript that addresses the points raised during the review process.

We would appreciate receiving your revised manuscript by Nov 08 2019 11:59PM. When
you are ready to submit your revision, log on to https://www.editorialmanager.com/pone/ and select the 'Submissions
Needing Revision' folder to locate your manuscript file.

If you would like to make changes to your financial disclosure, please include your
updated statement in your cover letter.

To enhance the reproducibility of your results, we recommend that if applicable you
deposit your laboratory protocols in protocols.io, where a protocol can be assigned
its own identifier (DOI) such that it can be cited independently in the future. For
instructions see: http://journals.plos.org/plosone/s/submission-guidelines#loc-laboratory-protocols

We look forward to receiving your revised manuscript.

Kind regards,

Ilan Harpaz-Rotem, PhD

Academic Editor

PLOS ONE

Journal Requirements:

2.Thank you for including your ethics statement: 

"All
participants gave written informed consent prior to inclusion. All studies described
in this manuscript were conducted in agreement with the ethical standards of the
research ethics committee of the Behavioural Science Institute, Radboud University
Nijmegen (ECSW2015-1105-310). "

Please amend your current ethics statement to confirm that your named ethics
committee Institutional Care and Use Committee (IACUC) specifically approved this
study.

For additional information about PLOS ONE ethical requirements for human subjects
research, please refer to https://www.google.com/url?q=http://journals.plos.org/plosone/s/submission-guidelines%23loc-human-subjects-research&source=gmail&ust=1569393054250000&usg=AFQjCNFL8nz6MHaTOwLHlxGpRzZ6yNEuEQ
http://journals.plos.org/plosone/s/submission-guidelines#loc-human-subjects-research
http://journals.plos.org/<wbr
/>plosone/s/submission-<wbr />guidelines#loc-human-subjects-<wbr
/>research. 

Additional Editor Comments (if provided):

The manuscript is interesting and carries important findings that need to be further
investigated and interpreted correctly. Both reviewers raised several significant
concerns that need to be addressed by the authors. Please respond carefully to all
the reviewers comments in the revised manuscript.

Reviewers' comments:

Reviewer's Responses to Questions

**Comments to the Author**

1. Is the manuscript technically sound, and do the data support the conclusions?

Reviewer #1: Yes

Reviewer #2: Yes

2. Has the statistical analysis been performed
appropriately and rigorously? 

Reviewer #1: Yes

Reviewer #2: Yes

3. Have the authors made all data underlying the
findings in their manuscript fully available?

Reviewer #1: Yes

Reviewer #2: Yes

4. Is the manuscript presented in an intelligible
fashion and written in standard English?

Reviewer #1: Yes

Reviewer #2: Yes

5. Review Comments to the Author

Reviewer #1: Intrusive memories are a hallmark feature of post-traumatic stress
disorder (PTSD), however, the frequency of intrusive memories can be reduced if
particular behavioral tasks (e.g., the computer game Tetris) are administered
following a traumatic event. Importantly, not all behavioral tasks reduce intrusive
memories following traumatic experiences and the mechanisms driving the effect are
currently unknown. Thus, work that outlines the boundary conditions that define
successful behavioral manipulations is critical. The paper under review seeks to
identify such parameters.

To investigate whether the visuospatial elements of behavioral tasks are the driving
force behind tasks that successfully reduce intrusive memories, this paper tested
the hypothesis that viewing attention-grabbing visuospatial stimuli during
consolidation of a trauma memory reduces the frequency of intrusive memories.
Intrusive memories were induced by an analogue trauma film within a healthy
population and were measured using a self-report diary in which participants tracked
post-film intrusive memories for one week. The paper reports null findings – the
frequency of intrusive memories was the same for control and experimental conditions
and the paper concludes that viewing visuospatial stimuli is not sufficient to
disrupt intrusive memories. The paper additionally explored whether reduced
frequency of intrusive memories is dependent on an interaction between the
visuospatial stimulation task and 1) working memory capacity, 2) working memory load
during the visuospatial task, and 3) concurrent trauma memory reactivation. It found
no interaction effects – the frequency of intrusive memories was the same across all
conditions. However, since the employed protocol did not mimic the same types of
visuospatial stimuli known to reduce intrusions, it is difficult to draw firm
conclusions from the work. For example, it may be the case that passively viewing a
pre-recorded Tetris game successfully reduces intrusive memories, even though the
stimuli used in the present study were not able to do so. Additionally, stimuli from
Tetris are often mentally replayed even after the game is complete. After the game,
players report seeing the world as a game of Tetris (e.g., visualizing falling
shapes or imagining how objects in their environment could be rotated to fit
together). One formal study noted intrusive, stereotypical, visual images during
sleep after Tetris play. This was also the case within amnesic patients when there
was no declarative memory of real-life gameplay (Stickgold et al., 2000, Science).
Thus, it may be the case that visuospatial stimuli are able to disrupt intrusive
memories, but only when the visuospatial stimuli are replayed in the days following
the trauma film. Mental replay that extends over days may have a more potent impact
on consolidation processes or more aggressively compete for resources needed for
consolidation. It would be helpful to know if mental images of the visuospatial
stimuli used in the present study induced replay during wakefulness and/or sleep,
and if so, how the frequency of such intrusions compare to that induced by passive
Tetris-watching and active Tetris gameplay.

As a separate comment, several of the studies cited in this paper are of
manipulations employed during a memory’s reconsolidation, not consolidation, phase.
Often times the paper does not clarify when findings from reconsolidation studies
are being used to justify the rationale of this consolidation study. Additionally,
the cited review articles are skewed toward manipulations that occur during trauma
movie watching, rather than during the consolidation period. Since there appear to
be fewer studies that have employed manipulations during the consolidation period,
it may be helpful to prioritize discussing those specific experiments over
referencing less-relevant work. It would specifically be useful to discuss
behavioral manipulations employed during the consolidation period that did not lead
to reductions in intrusion frequency. Currently, this is not thoroughly discussed,
but if included, could emphasize the nuance needed when studying manipulations as
such and clarify why the null findings of the present study are not completely at
odds with the currently available literature.

Reviewer #2: In this carefully-conducted piece, the authors attempt to replicate and
extend findings from a growing literature indicating that intervening with a
perceptual task after trauma-film viewing decreases numbers of intrusive thoughts
compared to no-task conditions. The authors attempt to manipulate the likely
components necessary to produce this effect: 1) distraction; and 2) working memory
load. Despite a well-conducted and adequately-powered study, the authors fail to
replicate past findings. Overall, I think this is an important contribution to the
literature that could be made even more impactful with the inclusion of one more
study, as below:

The authors demonstrate that intervention by neither working memory load manipulation
(controlled also for baseline working memory capacity) nor distraction is capable of
producing decreased frequency of intrusive memories after trauma film viewing. The
authors do an admirable job of discussing various components that could be important
for replication of these findings, but do not consider what may be the most obvious:
the need for both visually-arresting and working-memory-intensive tasks.

Without this control, the results of the study are interesting and worth reporting;
however, with it, the authors say something truly impactful about the mechanisms
necessary to achieve a decrease in intrusive memories. The authors should conduct
this experiment, or explain why they think the current results are able to speak to
the possibility that an interaction between these two components is necessary to
achieve the desired results.

6. PLOS authors have the option to publish the peer
review history of their article (what does this mean?). If published, this will
include your full peer review and any attached files.

If you choose “no”, your identity will remain anonymous but your review may still be
made public.

**Do you want your identity to be public for this peer review?** For
information about this choice, including consent withdrawal, please see our
Privacy Policy.

Reviewer #1: No

Reviewer #2: No

---

## [Author Response · Author response to Decision Letter 0]

20 Nov 2019

Journal Requirements:

Response: We checked style requirements and also updated the file names accordingly. 

2) Please amend your current ethics statement to confirm that your named ethics
committee Institutional Care and Use Committee (IACUC) specifically approved this
study. Once you have amended this/these statement(s) in the Methods section of the
manuscript, please add the same text to the “Ethics Statement” field of the
submission form (via “Edit Submission”).

Response: The ethics statement has been updated to clarify that: “All participants
gave written informed consent prior to inclusion. Study 1 and Study 2 were approved
by the research ethics committee of the Behavioural Science Institute, Radboud
University Nijmegen (ECSW2015-1105-310).”

Editor Comments 

1) The manuscript is interesting and carries important findings that need to be
further investigated and interpreted correctly. Both reviewers raised several
significant concerns that need to be addressed by the authors. Please respond
carefully to all the reviewers comments in the revised manuscript.

Response: We would like to thank the editor and the reviewers for their time and
effort and consideration of our manuscript. We carefully considered the comments and
suggestions that have been raised and revised the paper accordingly to address each
point.

Reviewer #1 

1) Intrusive memories are a hallmark feature of post-traumatic stress disorder
(PTSD), however, the frequency of intrusive memories can be reduced if particular
behavioral tasks (e.g., the computer game Tetris) are administered following a
traumatic event. Importantly, not all behavioral tasks reduce intrusive memories
following traumatic experiences and the mechanisms driving the effect are currently
unknown. Thus, work that outlines the boundary conditions that define successful
behavioral manipulations is critical. The paper under review seeks to identify such
parameters. To investigate whether the visuospatial elements of behavioral tasks are
the driving force behind tasks that successfully reduce intrusive memories, this
paper tested the hypothesis that viewing attention-grabbing visuospatial stimuli
during consolidation of a trauma memory reduces the frequency of intrusive memories.
Intrusive memories were induced by an analogue trauma film within a healthy
population and were measured using a self-report diary in which participants tracked
post-film intrusive memories for one week. The paper reports null findings – the
frequency of intrusive memories was the same for control and experimental conditions
and the paper concludes that viewing visuospatial stimuli is not sufficient to
disrupt intrusive memories. The paper additionally explored whether reduced
frequency of intrusive memories is dependent on an interaction between the
visuospatial stimulation task and 1) working memory capacity, 2) working memory load
during the visuospatial task, and 3) concurrent trauma memory reactivation. It found
no interaction effects – the frequency of intrusive memories was the same across all
conditions. 

Response: Thank you for this elaborate and accurate summary, as well as for sharing
our appreciation of research into the boundary conditions of behavioural
interventions against intrusive memories. 

2) However, since the employed protocol did not mimic the same types of visuospatial
stimuli known to reduce intrusions, it is difficult to draw firm conclusions from
the work. For example, it may be the case that passively viewing a pre-recorded
Tetris game successfully reduces intrusive memories, even though the stimuli used in
the present study were not able to do so. Additionally, stimuli from Tetris are
often mentally replayed even after the game is complete. After the game, players
report seeing the world as a game of Tetris (e.g., visualizing falling shapes or
imagining how objects in their environment could be rotated to fit together). One
formal study noted intrusive, stereotypical, visual images during sleep after Tetris
play. This was also the case within amnesic patients when there was no declarative
memory of real-life gameplay (Stickgold et al., 2000, Science). Thus, it may be the
case that visuospatial stimuli are able to disrupt intrusive memories, but only when
the visuospatial stimuli are replayed in the days following the trauma film. Mental
replay that extends over days may have a more potent impact on consolidation
processes or more aggressively compete for resources needed for consolidation. It
would be helpful to know if mental images of the visuospatial stimuli used in the
present study induced replay during wakefulness and/or sleep, and if so, how the
frequency of such intrusions compare to that induced by passive Tetris-watching and
active Tetris gameplay.

Response: We thank the reviewer for these thoughtful considerations and agree that
mental replay could be a highly relevant factor in the context of Tetris and
intrusions. Unfortunately, we do not know whether the visuospatial stimuli we used
induced any form of mental replay, since we did not include a task/measure of
spontaneous imagery involving these stimuli. Given this important comment, however,
we have extended the section addressing directions for future research accordingly
(pp.27-31). 

In particular, we now discuss that factors such as spatial planning, updating, or
coordination of motor movements could be controlled for more directly in future
studies, “e.g., by comparing actual Tetris gameplay to passive Tetris-watching”
(lines 640-642). 

Inspired by the reviewer’s comment, we also include the interesting idea that
interventions like Tetris could have effects after rather than during gameplay: “…
high levels of flow experience may result in altered experiences even after
gameplay, including continued replay, as well as intrusive visual images of the game
[58]. Speculatively, the beneficial effects that have been reported for Tetris may
thus depend on competition for visuospatial resources after, rather than during
gameplay. Therefore, an exciting avenue for future studies would be to explore if a
task’s ability to induce a flow experience, continued replay, and/or intrusive
imagery is required in order to interfere with intrusive memories of a trauma film”
(lines 648-655). 

3) As a separate comment, several of the studies cited in this paper are of
manipulations employed during a memory’s reconsolidation, not consolidation, phase.
Often times the paper does not clarify when findings from reconsolidation studies
are being used to justify the rationale of this consolidation study. Additionally,
the cited review articles are skewed toward manipulations that occur during trauma
movie watching, rather than during the consolidation period. Since there appear to
be fewer studies that have employed manipulations during the consolidation period,
it may be helpful to prioritize discussing those specific experiments over
referencing less-relevant work. It would specifically be useful to discuss
behavioral manipulations employed during the consolidation period that did not lead
to reductions in intrusion frequency. Currently, this is not thoroughly discussed,
but if included, could emphasize the nuance needed when studying manipulations as
such and clarify why the null findings of the present study are not completely at
odds with the currently available literature.

Response: We agree that a more specific focus on studies targeting the consolidation
phase following trauma film viewing is useful and have revised the introduction and
discussion sections accordingly. In particular, we now discuss a series of studies
that employed the game Tetris “immediately after a viewing a trauma film [7], after
30 min [e.g., 8, 9], or after 4 h [10]” (lines 51-53), and elaborate on their common
rationale that “newly formed memories are prone to retroactive interference for up
to several hours [11, 12]” (lines 56-57). In addition, we now emphasize more clearly
that our paper addresses interventions during the consolidation phase following
trauma film viewing throughout the entire manuscript and draw a more explicit
distinction with re-consolidation/re-encoding studies (e.g., lines 117-118; 372-376;
526-527). 

Furthermore, we have included two relevant recent papers (Asselbergs et al., 2018,
EJPT; van Schie et al., 2018, BRAT) that employed different visuospatial tasks
during the consolidation period following a trauma film. These papers reported null
findings across a series of studies and indeed help to clarify that the null finding
of our own experiments are not entirely at odds with the available literature. As we
now elaborate both in the introduction (lines 82-99) and the discussion (lines
584-600), the studies reported in both papers suggest that the effectiveness of
visuospatial tasks to reduce intrusive memories cannot be entirely attributed to
working memory taxation and/or individual differences in working memory capacity.
Next to our own work, these studies further emphasize the need for additional
research that we discuss in detail in the section “Future directions”. 

Reviewer #2 

1) In this carefully-conducted piece, the authors attempt to replicate and extend
findings from a growing literature indicating that intervening with a perceptual
task after trauma-film viewing decreases numbers of intrusive thoughts compared to
no-task conditions. The authors attempt to manipulate the likely components
necessary to produce this effect: 1) distraction; and 2) working memory load.
Despite a well-conducted and adequately-powered study, the authors fail to replicate
past findings. 

Response: Thank you for this summary and for the encouraging words concerning our
methodology.

2) Overall, I think this is an important contribution to the literature that could be
made even more impactful with the inclusion of one more study, as below:

The authors demonstrate that intervention by neither working memory load manipulation
(controlled also for baseline working memory capacity) nor distraction is capable of
producing decreased frequency of intrusive memories after trauma film viewing. The
authors do an admirable job of discussing various components that could be important
for replication of these findings, but do not consider what may be the most obvious:
the need for both visually-arresting and working-memory-intensive tasks.

Without this control, the results of the study are interesting and worth reporting;
however, with it, the authors say something truly impactful about the mechanisms
necessary to achieve a decrease in intrusive memories. The authors should conduct
this experiment, or explain why they think the current results are able to speak to
the possibility that an interaction between these two components is necessary to
achieve the desired results.

Response: We fully agree that based on Study 1, one could expect visually arresting
stimulation to be effective only if it is combined with intensive working memory
load. Indeed, the design of Study 2 was based on this reasoning: in both active
conditions, participants were exposed to the visually arresting stimuli whilst
executing the WM-intensive tasks involving memorizing and imagining. 

Strictly speaking, it is correct that Study 2 did not formally test the interaction
between high/low WM load on the one hand, and high/low visually arresting stimuli on
the other hand. However, Study 1 tested low WM load combined with H-VA vs. L-VA
stimuli, while Study 2 addressed high WM load combined with H-VA stimuli.
Importantly, both studies included no-task control groups, and none of the active
interventions differed from this common reference group in either study. In
addition, there were no linear associations between WMC (Study 1) or task engagement
(Study 2) with intrusive memories. In light of the absence of condition effects and
associations between the critical concepts, we believe there to be little reason to
expect an interaction effect in a full factorial study.

However, following this valuable comment, we adapted parts of the manuscript
accordingly. Specifically, we have revised our discussion of Study 2 on p.28,
clarifying that Study 2 indeed addressed the combination of high load on WM
resources and complex visuospatial stimuli. In addition, we have now extended the
discussion of relevant null findings in the literature (also see Reviewer 1, point
3). Based on the review comments, we have also extended the discussion of avenues
for future research that we believe to be the most promising to unravel when, why,
and how visuospatial tasks can interfere with intrusive memories (pp.27-31).

---

## [Decision Letter · Decision Letter 1]

15 Jan 2020

Arresting visuospatial stimulation is insufficient to disrupt analogue traumatic
intrusions

PONE-D-19-21303R1

Dear Dr.
Thomas
Meyer,

We are pleased to inform you that your manuscript has been judged scientifically
suitable for publication and will be formally accepted for publication once it
complies with all outstanding technical requirements.

With kind regards,

Ilan Harpaz-Rotem, PhD

Academic Editor

PLOS ONE

Additional Editor Comments (optional):

Reviewers' comments:

Reviewer's Responses to Questions

**Comments to the Author**

1. If the authors have adequately addressed your comments raised in a previous round
of review and you feel that this manuscript is now acceptable for publication, you
may indicate that here to bypass the “Comments to the Author” section, enter your
conflict of interest statement in the “Confidential to Editor” section, and submit
your "Accept" recommendation.

Reviewer #2: All comments have been addressed

2. Is the manuscript technically sound, and do the data
support the conclusions?

Reviewer #2: Yes

3. Has the statistical analysis been performed
appropriately and rigorously? 

Reviewer #2: Yes

4. Have the authors made all data underlying the
findings in their manuscript fully available?

Reviewer #2: Yes

5. Is the manuscript presented in an intelligible
fashion and written in standard English?

Reviewer #2: Yes

6. Review Comments to the Author

Reviewer #2: Despite the fact that the authors did not perform the experiment
requested, they clarify their reasoning and add evidence from the literature to
support their conclusions.

7. PLOS authors have the option to publish the peer
review history of their article (what does this mean?). If published, this will
include your full peer review and any attached files.

If you choose “no”, your identity will remain anonymous but your review may still be
made public.

**Do you want your identity to be public for this peer review?** For
information about this choice, including consent withdrawal, please see our
Privacy Policy.

Reviewer #2: Yes: Albert R. Powers III, MD, PhD

---

## [Editor Report · Acceptance letter]

17 Jan 2020

PONE-D-19-21303R1 

Arresting visuospatial stimulation is insufficient to disrupt analogue traumatic
intrusions 

Dear Dr. Meyer:

I am pleased to inform you that your manuscript has been deemed suitable for
publication in PLOS ONE. Congratulations! Your manuscript is now with our production
department. 

With kind regards,

on behalf of

Dr. Ilan Harpaz-Rotem 

Academic Editor

PLOS ONE